# Caring for Tuberculosis Patients: Understanding the Plight of Nurses at a Regional Hospital in Limpopo Province, South Africa

**DOI:** 10.3390/ijerph16244977

**Published:** 2019-12-07

**Authors:** Hulisani Matakanye, Dorah U. Ramathuba, Augustine K. Tugli

**Affiliations:** Faculty of Health Science, Department of Public Health, University of Venda, Thohoyandou 0950, South Africa; dorah.ramathuba@univen.ac.za (D.U.R.); Tugli.augustine@univen.ac.za (A.K.T.)

**Keywords:** caring, nurses, patients, plight, tuberculosis, understanding

## Abstract

Tuberculosis (TB) is a disease which is caused by a relatively large, non-motile, rod-shaped pathogen called *Mycobacterium tuberculosis*. TB is a major cause of illness and death worldwide, especially in Asia and Africa. Despite the fact that TB is a curable illness, the tragedy is that TB remains the biggest killer in the world as a single pathogen. The aim of this study was to determine the experiences of nurses caring for TB patients at a regional hospital in Limpopo Province, South Africa. Qualitative, exploratory, and descriptive designs were used. A non-probability purposive sampling method was used to select the participants. The personal experiences of six nurses with more than five years’ experience caring for TB patients at a regional hospital were explored, and it was guided by data saturation. Data were collected through in-depth individual interviews. Data were analyzed using Colaizzi’s method. Trustworthiness was ensured and ethical considerations were observed in this study. The research findings revealed six major themes from the raw data: challenges of the working environment, problems impacting on the quality of nursing care, fear, anxiety, stress and risk of contracting infection, nurses’ perceptions towards patients, support structure available in the hospital, and support needs for the nurses. Therefore, there is an urgent need to address the challenges experienced by nurses caring for communicable diseases through provision of a positive practice work environment.

## 1. Introduction

Tuberculosis (TB) is a major cause of illness and death worldwide, especially in Asia and Africa. Despite the fact that TB is a curable illness, the tragedy is that TB poses a grave danger to healthcare workers including nurses because of their exposure to TB patients in health facilities. According to the World Health Organization [1], TB remains the biggest killer in the world as a single pathogen, and the healthcare systems are being overwhelmed by the increasing number of TB cases. 

It is estimated that one-third of the world’s population is infected with TB which is responsible for 2–3 million deaths annually [1]. A study conducted by Valjee and van Dyk [2] in South Africa, reported that nurses caring for patients living with TB related illnesses, including HIV/AIDS expressed anxiety, helplessness, and vulnerability.

Nurses play a major role in caring for patients infected with different infectious diseases. Nurses in high TB burden settings are at higher risk of developing latent tuberculosis infection (LTBI) when compared to the general population, due to their exposure to the large number of smear-positive TB cases managed at hospitals or healthcare facilities [3]. Working in a high-risk TB setting can lead to harrowing experiences for healthcare, especially where health facilities are poorly equipped and managed. Health Care Workers (HCWs), especially nurses, have higher rates of latent and active TB than the general population due to persistent occupational TB exposure, particularly in settings where there is a high prevalence of undiagnosed TB in healthcare facilities and TB infection control programmes are absent or poorly implemented [4]. 

In the Limpopo Province of South Africa, TB is ranked the fourth among the major causes of death [5]. In 2009, the Limpopo Provincial TB Annual Report [6] stated that the number of TB cases in Vhembe district was 2194. In 2012, Vhembe district had the highest number of TB patients in the Limpopo Province [7]. Tshitangano et al. [8], further indicated that TB infection control plans were not available at hospitals in Vhembe district and this contributed to the high number of TB patients. Infection control requires both human and material resources for maintenance and support [8]. In line with this assertion, Gursimrat [9] argues that one of the major challenges to control TB infection includes poor primary healthcare infrastructure. In addition, shortage of space and beds in the wards hinder separation of Multi-Drug Resistant TB (MDR-TB) and TB patients in healthcare facilities and these increases risk of contracting infection in the wards [10]. Healthcare-associated TB has become a major occupational hazard for healthcare workers. It was therefore imperative to explore and describe the experiences of nurses caring for tuberculosis patients at a regional hospital in Vhembe district, Limpopo Province.

## 2. Methods 

### 2.1. Study Design

The study used a qualitative, exploratory, and descriptive design to explore and describe the experiences of nurses caring for TB patients at regional hospital. The researcher wanted to uncover the personal experiences of nurses and the meanings that they attached to the events, processes, and structures of their lives in the TB wards. Therefore, an exploratory and descriptive design was the most suitable design.

### 2.2. Study Site

This study was conducted at a regional hospital in Vhembe district in Limpopo Province, South Africa between 2014 and 2015. The hospital is a public healthcare facility situated in Thohoyandou, Limpopo province of South Africa. It is the only regional hospital in Vhembe district of Limpopo province with two TB wards and 20 beds. It is a referral hospital for seven community hospitals in Vhembe district. According to Wolmarans and Asia [11], the total population is about 1,294,722, comprising of Vha-venda, Va-tsonga, and Sotho speaking people. 

### 2.3. Study Population

The study the target population comprised of all nursing staff in all TB wards caring for TB patients at a regional hospital in Vhembe District in Limpopo Province at the time of this study. The total population of nurses working in the TB ward was 22 nurses. 

### 2.4. Sampling Procedure

The study used a non-probability purposive sampling method, since it involves selection of individuals for participation based on their knowledge of a phenomenon that is being studied. It was believed that those participants would provide the researcher with the rich data needed to gain insight and discover new meaning in an area of study. The inclusion criteria were all nurses with five years’ experience caring for TB patients since it was believed that they would provide information-rich data.

### 2.5. Sample Size Determination

The estimated sample for the interview was 15 as they met inclusion criteria and 10 participants agreed and consented to participate, and data saturation was reached with six participants. Interviews were terminated when data saturation was reached, that was when information was repeated and when the researcher probed, rephrased questions, and requested clarity but the information kept repeating itself. 

### 2.6. Data Collection Procedure

Data was collected through in-depth individual interviews. All participants who signed a written consent form before data collection were included in this study and were interviewed. The researcher explained the purpose of the study to each participant before interviewing them. Participants were assured that confidentiality will be maintained. The interview dates and times were arranged with participants prior to data collection date. Data collection took place between April–July 2015. The interviews were conducted early in the morning as suggested by participants before they could start with their duties to avoid distracting them from their normal ward routine work. Separate interviews with six participants took place in a private office within TB wards at the regional hospital, and each lasted approximately 30 min. All interviews were conducted by the researcher. All interviews started with an opening question: *“What is your experience when nursing patients who suffer from TB?”*

The research participants freely responded to open-ended questions in narrative form using their own words, thus sharing their own perspectives with the researcher. Questions were not planned in an inflexible manner. The questions were not asked in a pre-arranged sequence, but the researcher ensured that all relevant topics were covered and that the research focus was kept in mind. The researcher also asked probing questions to guide participants to elaborate further upon their responses where additional information were required or where unclear answers required more clarity. This resulted in gaining in-depth accounts about participants’ experiences while caring for TB patients. Data was collected by means of audio-recordings, field notes, and in-depth interviews. 

### 2.7. Data Management and Analysis

Data was stored in a password-protected computer. Access to the database was restricted to the researcher and supervisors only. Data was stored as per university’s protocols. Any identifiable information that was collected remained confidential and was only accessible to the researcher and supervisors. Data was analyzed in groups not individually to avoid identifying the participants by their responses. Qualitative data analysis always takes place concurrently with data collection. Therefore, the researcher attempted to gather, manage, and interpret a growing bulk of data simultaneously. In this study, the audio-recorded interviews were transcribed and coded immediately after data collection. Data was analyzed using Colaizzi’s [12] methods which comprise the following seven steps:Each research participant’s verbatim transcript was read to acquire a sense of the whole.Significant statements and phrases pertaining to the phenomenon being studied were extracted from each transcript.Meanings were formulated from the significant statements.Meanings were organized into themes, and themes evolved into theme clusters and eventually into theme categories.These results were integrated into a rich and exhaustive description of the lived experience.The essential structure of the phenomenon was uncovered.Validation was sought from the research participants to compare the researcher’s descriptive results with their lived experiences.

### 2.8. Ethical Consideration

The proposal was submitted and presented to the School of Health Science and University Higher Degrees Committee (UHDC) and ethical clearance was granted (SHS/15/PH/07/1604). Permission to conduct the study was obtained from the Limpopo Provincial Department of Health and Vhembe District Department of Health, Nursing Service Manager, and Chief Executive Officer of the regional hospital.

Consent forms were given to every participant to complete. The nature of research was described to the participants of the study and they were informed of their right to refuse to participate, or to withdraw from participating if they felt that they could not continue. The participants were also informed and assured that the information they provided would not be used against them or shared with other people but would be reported as findings of the study.

Anonymity was also ensured in which the participants did not write down their names or any personal identification. This allowed the participation to be anonymous. The researcher respected the choices and agreements made with the participants. The initial agreement was not changed without the knowledge of the participants. The participants were not victimized for refusing to participate in the study.

## 3. Results

After data organization and analysis, six major themes were developed from the findings of this study and sub-themes were also formulated (see Table 1). Participants shared the same experiences with regards to caring for TB patients at Tshilidzini Hospital.

## 4. Interpretation of Data

Six major themes emerged after extensive data organization and analysis. These themes were developed from the comments that were made by the participants during data collection. A discussion of findings follows the presentation of each theme, as it is asserted that integrating the findings and the discussion is an appropriate method for encapsulating the essence of the phenomenon under investigation. Existing literature was searched and used to support the findings of this study.

### 4.1. Changes in the Working Environment

Participants articulated that the hospital has a poor, aging infrastructure which is in dire need of refurbishment. TB wards were not structured to the standard that is suitable for TB wards and this does not induce the productivity of nurses in the wards. The following sub-themes were highlighted in the context of working environment, physical environment, and practice environment.

#### 4.1.1. Physical Environment

A good working environment can facilitate better service and reduce workload. A disorganized working environment impairs the health center teams. Infrastructure includes attention to space, ventilation, comfort, and rational layout that facilitates links within the healthcare settings. The workplace has an impact on preventing TB transmission; it serves as the first line of defense for preventing the spread of TB in the healthcare settings. Participants expressed similar sentiments about poor infrastructure.

Nurse (N1) said: “the hospital infrastructure is very old and deteriorating, the wards’ layout is not conducive for TB wards, it has small windows that fuel bacterial concentration.”

Another nurse (N2) added that: “the working environment is poor since we sit in the wards with patients all day long. During lunch time we sit in a small room within the TB wards, which accommodates less than five people and the door doesn’t even close.”

Nurse (N4) said: “all patients are accommodated in the same wards, whether they are extensively drug-resistant TB (XDR-TB) or MRD or TB patients, hmmm the wards layout is not proper. We have to sit all day with patients and told that patients who are under treatment are not infectious (emotional).”

According to the findings of this study, physical environment was poor and TB ward designs were not conducive. A study that was done by Kieft et al. [13] indicated that improved hospital design can help reduce staff stress and fatigue and increase effectiveness in delivering care, improve patient safety, reduce patient and family stress, and improve outcomes and the overall healthcare quality. The findings of this study concur with the study that was done by Brophy [14], which indicated that the nurses were concerned with their occupational environment and voiced concern about the deteriorating hospital infrastructure.

#### 4.1.2. Poor Positive Practice Environment

Positive practice environment is the setting that supports excellence and decent work. To ensure positive practice environment in the healthcare setting, it is important to consider health, safety, and personal well-being of the staff. These improve productivity and performance of the staff, it also benefits healthcare consumers, and leads to positive health indicators for the government. 

Nurse (N3) said: “Sometimes we are supplied with fake N95 masks or required to use surgical masks. We only get 21 masks per week, to share with social workers, doctors and other staff that come to the TB wards. We do not get enough gloves, and we do not have aprons. We sometimes lack soap to wash our hands after coming into contact with TB patients, and we also have limited washing basins. In addition, the windows are too small and there is poor airflow in the wards. UV lights are not regularly service.”

Nurse (N1) added that: “This whole thing (emotional) of us being given fake masks results in negative practice for patient-care since we are afraid to contract TB. Each time we look at the patients we get scared of being in the same situation, hmmm, there are more negative thoughts in TB wards than positive ones.”

The findings of this study revealed that there was a more negative practice environment in the TB wards than a positive one. According to the study that was done by Washeya [15], it is important that the working environment enhances productivity. The findings of this study indicated that there was a poor practice environment in the TB wards. There was also an inadequate supply of personal protective materials. This leads to negative practice by the nurses, since they are afraid to contract diseases. The findings of the present study concur with those of a study that was done by Mametja [16], which indicated that there was a more negative practice environment in HIV/AIDS wards which affected the quality of nursing care in the public hospital. 

### 4.2. Problem Impacting on the Quality of Nursing Care

The main reason why patients should be admitted at the hospital is that they should be provided with quality care by healthcare professionals to recover well. However, there are challenges that impact quality of care. The following sub-themes will be discussed; adherence to treatment, delay in diagnosis, lack of equipment and working resources, and lack of enough skills and in-service training. 

#### 4.2.1. Adherence to Treatment

Patients default on treatment for different reasons, such as no longer feeling sick, lack of knowledge, personal or cultural beliefs, lack of access to healthcare facilities, lack of motivation, and poor relationship with healthcare workers. Non-adherence is one element that impacts negatively on the quality of nursing care. Below are the comments made by the participants. 

Nurse (N4) indicated that: “The major problem that we have is that (emotional) our hospital is a rural-based and most patients are from the poor backgrounds; so, they always default their treatment for various reasons, such; continue to receive TB grant, bad side effects of medication, social problems and long distances to the clinics where they should collect their medications; lack of money to buy food since TB drugs make them eat more.”

Nurse (N2) added that: “There is a high TB prevalence in surrounding rural areas. Some of them default because they go to traditional healers and the healers tell them that they have been bewitched; they are not really sick. When those patients come back to the hospital their condition will be bad and most of them do not survive a week. Some patients continue to smoke and drink alcohol and worsen their conditions as they fail to adhere to their treatment.”

The findings of this study revealed that patients continued to default treatment for different reasons, such as experiencing side effects, alcohol abuse, cultural and religious beliefs, and lack of access to healthcare facilities. Some default because they wanted to continue receiving the TB grant. The findings of this study concur with those of a study that was done by Herrero et al. [17], which listed the barriers contributing to poor TB treatment compliance, such as communication difficulties, low literacy level, inadequate knowledge and low awareness of TB disease, patients’ attitudes and beliefs in treatment efficacy, depression and other psychiatric illness, alcohol and substance abuse, unstable living conditions, negative health provider attitudes, stigma and discrimination, overcrowding, and access to medicine. 

#### 4.2.2. Delays in Diagnosis

Tuberculosis control can be effectively achieved if individuals with the disease receive adequate and timely treatment. Early diagnosis and prompt effective therapy form the key elements of the tuberculosis control programme. Delay in diagnosis results in increased infectivity in the community.

Nurse (N3) said: “Most of the patient are critically ill. Most of them report to the hospital when their conditions have deteriorated. They first go to traditional healers for help.”

Nurse (N1) indicated that: “Even when patients are diagnosed with TB they will still go to traditional healers for help because they do not believe they are ill; rather they believe that they have been bewitched. Some of them won’t even disclose their status to their family members because of fear of being stigmatized. Many people in deep rural areas lack understanding.”

According to the findings of the present study delays in diagnosis were due to the following factors: fear of stigma, patients’ dissatisfaction with service from private healthcare providers, lack of knowledge, patients’ beliefs, and patients choosing to consult traditional healers rather than healthcare facilities. The findings of this study concur with the study conducted by Mbuthia et al. [18], many people in Africa visit traditional healers prior or concurrently with formal healthcare services. Delays in TB diagnosis is common in many regions worldwide, due to patient related factors, such as stigma, lack of information, dissatisfaction with TB service, inaccessibility of treatment, or provider-related factors such as diagnosis delay, knowledge and skills of healthcare workers, and inadequate infrastructure.

#### 4.2.3. Lack of Equipment and Working Resources

Improving the productivity and performance of health workers to ensure that health interventions are efficiently delivered is important in the healthcare setting. Human resources for health, consisting of clinical and non-clinical staff, are the most important assets of health systems. The performance of a health organization depends on the knowledge, skills, and motivation of individuals. It is important for employers to provide suitable working resources to ensure that the performances of employees meet the desired standards. 

Nurse (N5) said: “we do not have essential equipment in the TB wards: UV lights, fans and air-conditioner are not working properly hmmm. I do not think they have ever been serviced since they were installed. Airflow in the wards is very poor because the wards are too small.”

Another (Nurse 4) added: “There is a poor supply of N95 masks (sad) and it puts us in danger of contracting infection. But we are surviving by the grace of God. We are told that the hospital doesn’t have enough money to buy masks.”

According to the findings of this study, there is a shortage of equipment and working resources were impacting the quality of health in the TB wards. Participants appeared to be scared and were discouraged by the poor supply of resources in the wards. There was poor supply of N95 respiratory masks, UV lights were not regularly serviced, and there were no fans and air conditioners in the wards. The wards had small windows. There was also poor hygiene practice since there was a poor supply of detergents and hand washing soap. A study done by Mosadeghrad [19] concludes that inadequate or non-available facilities, equipment, and resources, especially basic resources to provide services such as water, steam, and electricity, hindered the provision of quality healthcare, proper conduct of tests, therapies, investigations, and surgery. Another study done by Reilly [20], concluded that patients are almost twice as likely to die in some hospitals because they lack the proper equipment and resources.

#### 4.2.4. Lack of Sufficient Skills and In-Service Training

Healthcare delivery can be highly labor-intensive and frustrating if the healthcare worker is not well trained. The quality, efficiency, and equity of services are dependent on the availability of skilled and competent health professionals. It is essential that health workers are appropriately trained to deliver the required services according to set standards. 

Nurse (N5) indicated that: “our biggest challenge in the wards, is that we only have one nurse who is knowledgeable about TB. She is the only specialist in the wards; the rest of us know TB a bit because we read about it and from the experience that we have as nurses, even though some of us have attended workshops but I do not recall having to go for training. TB patients undergo different stages and it is important for us to understand them. Hmm some of the behaviors that we see in the wards shock us and we do not understand why patients could behave like they are insane.”

A study by ten Hoeve et al. [21], found that one of the independent functions of a nurse is education. According to a study that was done by WHO [1], healthcare professionals need to seek up-to-date scientific knowledge from national, and international, academic and research institutions, including professional associations. The findings of this study revealed that there was a lack of skills and in-service training in the wards. This finding concurs with those of a study conducted by Yang et al. [22], which indicated various barriers contributing to poor TB treatment, such as communication difficulties, low literacy level, inadequate knowledge, and low awareness of TB disease.

### 4.3. Fear, Anxiety, Stress, and Risk of Contracting Infection

TB is a contagious disease and it is usually spread through the air by droplet nuclei. Transmission generally occurs indoors, in dark and poorly ventilated spaces. The following sub-themes will be discussed under fear, anxiety, stress, and risk of contracting infection: Exposure to risk, fear to contact patients, and patients spreading disease.

#### 4.3.1. Exposure to Risk

TB patients are likely to transmit the disease to the healthcare workers since TB is an airborne disease. This aspect of occupational risk is largely understudied, and preventive measures are frequently not in place. This problem is more in the low-to middle-income countries, due to increased prevalence of TB and lack of effective control programmes.

Nurse (N6) said: “we are exposed to a great risk of being infected by TB, MDR-TB and XDR-TB because we do not have personal protective materials in place and hmmm the hospital also has poor infection control measures.”

Another Nurse (N4) added: “the gloves that we get are not enough as well and we do not have aprons to cover ourselves. We put ourselves in a great danger of contracting infection, sometimes I get scared if I will not infect my family because the same clothes that I use at work is the ones that I go with at home and my children will come running when I am back from work I am scared to give them a hug because I do not know if my clothes are contaminated.”

According to the findings of this study it was revealed that nurses feared contracting TB infection. Even though participants were provided with personal protective materials they were of poor quality. Participants seemed to be more concerned about their safety in the wards and they appeared to be more stressed and emotional about the danger that they are exposed to in the wards. A study done by Engelbrecht et al. [23] reveals that nurses consider themselves more vulnerable and at high risk of contracting infection.

#### 4.3.2. Fear of Contacting Patients

TB is a major occupational hazard for Health care workers (HCWs) worldwide. The transmission of drug-sensitive and drug-resistant strains of *Mycobacterium tuberculosis* occurs through infected droplets aerosolized by patients with active pulmonary TB. The transmission risk to HCWs is highest when patients have unrecognized TB or are receiving inappropriate treatment. However, many other factors influence the risk of transmission and progression to active disease, including healthcare setting, occupational category, individual susceptibility/immune status, and the adequacy of TB infection control measures.

Nurse (N3) said that: “Most of the patients who are admitted in our wards are being transferred from other wards due to their critical conditions. When some wards such as medical wards realize that TB patients’ condition has deteriorated, they transfer them to our wards, and by then you will find that they cannot do anything on their own. They need us to feed them, bath them and take them to the toilet.”

Nurse (N4) said: “Myself, I feel like I am putting myself in danger to be honest. I am scared to come into contact with those patients because most of them are rude and dangerous.”

The findings of this study show that nurses were scared to come into contact with TB patients. As a result, they tend to spend less time with them. Nurses tend to neglect patients because of fear of getting infected and they provide less nursing care to those patients. According to the study done by Valjee and van Dyk [2], healthcare workers express a sense of physical and mental exhaustion and stress from dealing with HIV/AIDS patients that need serious attention and care. The findings of this study concur with those of a study by Wyzgowski et al. [24] which showed that HIV/AIDS patients often tried to expose the nurses and their relatives to their blood and fluids on purpose. 

#### 4.3.3. Spread of the Disease

TB patients can release tiny particles containing *Mycobacterium tuberculosis* into the air by coughing, sneezing, laughing, or singing. These particles are called air droplet nuclei. They are invisible to the naked eye. Droplet nuclei can remain airborne in room air for many hours, until they are removed by natural or mechanical ventilation. For TB to spread, there must be a person with TB disease who produces the TB bacilli, and another person who inhales the droplet nuclei containing the bacilli. Although TB is not usually spread by brief contact, anyone who is in close proximity with an infectious person is at risk of getting infected.

Nurse (N1) indicated that: “It is sad the way TB is spreading in our communities, we have a record of patients who come from one family or who are related. They decide to infect others or share their sputum just for them to get a TB grant. When they realize that they are recovering, they default for the social grant to continue. Most of those patients, when they come back to the hospitals, they will be MDR or XDR; they do not survive. Some MDR or XDR TB patients infect one another in the TB ward.”

Nurse (N6) had this to say: “Patients also spread disease when they use public transport. They go to church and when they are in shopping complex. Each time I hear of TB outbreak in the communities I get stressed because I see large number of people who are still going to die of TB. We feel vulnerable because in the whole community we are the first people in the line of danger.”

According to the findings of the present study, TB patients spread TB in the wards and outside. The nurses indicated that they feel vulnerable, since they are at great risk of contracting TB. The nurses indicated that patients continue to spread infection, even when they are outside TB wards and that contributes to an outbreak of TB in rural areas. The findings of this study concur with those of the study done by Arjun [25], which concluded that TB patients are a risk to community members because when they go to shopping centers and places where there are large numbers of people, such as social gatherings and church services they become a danger to the people.

### 4.4. Perceptions of TB Patients

Nurses provide nursing care to the patients during the period of their hospitalization. The way nurses perceive patients matters; the nurses play a major role in the patients’ recovery. The following sub-themes will be discussed under nurses’ perceptions of their work; feelings and nurses’ emotions towards patients.

#### Feelings and Emotions of Nurses Towards Patients

Nurses who care for dying patients are under pressure emotionally because of their beliefs and values about death as well as the emotions and reactions of the patients and their families. Despite TB being a curable disease, in almost all new cases, professionals such as psychologists, anthropologists, sociologists, and TB analysts have repeatedly indicated that talking about the disease still causes discomfort and unease in the population, especially within the poorer communities [26]. TB is surrounded by intense grief, with implications to different spheres of life, including social relationships. This is due to long-standing negative representations about this disease, which result in stigma and discrimination. Healthcare professionals believe besides TB’s ability to impact patients’ lives and causes death, it also has strong emotional impact on the life of a patient since it evokes disability, impotence, and self-discrimination. 

Nurse (N6) reiterated that: “Most of our patients are very ill and their conditions are not good at all, you look at the person and see that chances are he/she might die, even though the hospital doesn’t have enough resources for us to do our job, but (emotional) we can’t just sit here and look we need to help those patients they become our responsibilities, we rather endanger our lives in order to save lives. I remember we have several cases where two patients from one family will be admitted and their beds will be next to each other, it always happens that one will die and the other one will remain, it will be so touching some even ask if he/she will be fine, to avoid frustrating the patient we always request that he/she be transferred to another hospital.”

Nurse (N3) indicated that: “Every time when I get to the wards I look at those kids some of them are very young and I just wish they would recover, they become so close to us like they are our own children or family members, we always encourage them to take medication and injections even when it is painful we help talk to them. Most of patients have side effects and they will vomit after taking medication and we always have to go all out and look for the food because they will be hungry.”

According to the findings of this study, nurses experience stress, frustration, and pain due to the conditions of TB patients and they perceive their job as one of the hardest since it deals with emotions. Nurses always go to an extent of risking their lives in order to save lives. When they fail, they always feel bad and become emotionally frustrated. This study shows that, there is a high death rate in the TB wards of Tshilidzini Hospital, and when a patient dies, nurses feel sad and hurt. The findings of this study agree with those of the study by Khalid et al. [27], which showed that participants also experienced a variety of emotions when nursing patients with Severe Acute Respiratory Syndrome, and not knowing what they were facing.

### 4.5. Support Structure Available in the Hospital

Due to the nature of work that nurses provide to the patients, it is important that they receive special support from their management and colleagues in order for them to offer quality service. The following sub-themes will be discussed under support system available in the hospital; support from the managers, support from the psychologist and appreciation of staff. 

#### 4.5.1. Support from the Managers

Nursing managers are widely accepted as the most influential force in staff satisfaction and retention because of their role in work environments. Hospital management protects and provides staff with the resources to do their job. Within the clinical settings, managers are consistent in their presence; uniquely positioned with a front-row view of the intricacies of nurse-patient, nurse-physician, and nurse-interdisciplinary team dynamics. Hospitals, together with nursing managers, are expected to oversee the daily demands of unit operations while developing an environment that fosters nursing excellence and promotes an engaged nursing staff. On the other hand, nursing managers are the vital link between hospital senior executives and direct care nurses. Good management makes a hospital meet its complex target. Below are the comments made by the participants:

Nurse (N5) commented positively saying that: “we do receive full support that we need in the wards from the managers. But we are happy of the fact that the hospital CEO does support us, and the nursing service manager gives us support.”

Nurse (N6) also commented that: “Hmmm we believe that for now we just need a special support from the management and all TB programme leaders, to give us more knowledge and skills on TB.”

Nurse (N3) indicated that: “I believe that as a nurse it is more crucial that we have meetings with the doctors and other management staff.”

According to the findings of the present study even though the support structure from the management is available, there is a need for special support for TB patients. Lack of support from the management confuses the nursing staff and causes staff to lose interest in their job. The findings of this study concur with those of the study by Manyisa and van Aswegen [28], which revealed that South African hospitals’ working environments lack resources and managerial support.

#### 4.5.2. Support from Psychologists

Nursing has long been considered as one of the most stressful professions. Stress in nursing is attributed largely to the physical labor, suffering, and emotional demands of patients and families; long working hours, shift work, and interpersonal relationships. Sophisticated healthcare technologies, budget cuts, increasing workload, and constant organizational changes in some healthcare environments appear to be the factors that have increased stress among nurses. Nurses seem to be overexposed to a range of psychosocial stressors. As a result, it is important that they receive psychological support regularly in order for them to cope in their working environment and remain productive. Below are the comments made by the participants:

Nurse (N2) indicated that: “The most painful and traumatizing thing in the TB wards is that patients die in large numbers and it is so stressing to see people dying just before you every day. We really need clinical psychologists to come and render counselling, but all they do is to schedule an appointment with us and never come to talk to us. We do not get even debriefing sessions. As a result, we feel that we are losing it, at times we just can’t cope.”

The findings of this study revealed that nurses do not receive personal psychological support. Even though the support structure is there in the hospital, their needs are not considered and attended to. A study by Mametja [16] reported concerns that were raised by nurses, that they needed counselling, not only when they have contracted the disease, but for them to cope with the demands of the high number of patients who no longer recover from the illness, but die as the result of HIV/AIDS. The findings of the present study agree with those of the study by Arjun [25], which concluded that nurses caring for MDR-TB patients did not receive psychological support from the clinical psychologists when they need it; neither was there a functional employee assistance programme available at the institution.

#### 4.5.3. Appreciation of Staff

Hospitalized patients require more than a dozen daily medications that are administered at various hours and in multiple ways. Although physicians have the responsibilities to diagnose infections and prescribe medications for patients, nurses are responsible for actively caring, feeding, bathing, and dispensing medication for hospitalized patients throughout the day. Nurses are on the front line of fighting all kinds of diseases in the world, and they risk their lives to provide quality healthcare services in the hospital.

Nurse (N2) showed concern and said: “the hospital doesn’t even appreciate us for the good work that we are doing by risking our lives daily. We do not even get a danger allowance in case we contract infection in the wards. Instead we are being told that if we are found with TB we would be moved to another ward which is not a TB ward.”

Nurse (N1) further indicated that: “we do not have TB awareness campaign, even just for a day, where the management just appreciates us for the great work that we are doing. At times I even consider other opportunities. Even the families of patients blame us for everything that goes wrong with the patients they do not appreciate the effort we put to care for the patients.” 

According to the findings of the present study TB nurses feel unappreciated for what they are doing. TB nurses are on the frontline of the fight against the highly infectious diseases that affect many people. In spite of this, they do not receive a danger allowance and the hospital does not celebrate TB day as a way of appreciating TB nurses as pioneers in the hospitals who risk their lives in order to fight TB disease. In support of these findings, Loghmani et al. [29] reported that the participants’ experiences were that the families believed that nurses did not do their best for the patients and that they abused the patients. The findings of this study concur with those of the study by Sodeify et al. [30], which concluded that South African nurses were not well supported, appreciated, recognized, and well-rewarded; unlike in other countries since they provide quality service to patients under very difficult circumstances with limited resources. This is why they leave South Africa to work in other countries. 

### 4.6. Support Needs for the Nurses

Evidence indicates that nurses have a heavy workload and they work under pressure on a daily basis. They are expected to have high levels of independence, well-developed problem-solving abilities, and leadership responsibilities. Regardless of all the skills they have acquired they still express needs that are not met by the healthcare managements. The following sub-themes were developed under support needs for the nurses: good ventilated working environment, good infection control and prevention measurements, and special support from management.

#### 4.6.1. Good Ventilated Working Environment

TB is transmitted through airborne routes. Poorly designed or overcrowded healthcare facilities play an important role in TB transmission and also increases the chances of cross-infection between patients and healthcare workers. HCWs are essential in the fight against TB infection and they should be protected. A good ventilated working environment should have natural, mechanical, and negative pressure ventilation to reduce the risk of spreading TB infection in the wards. Natural ventilation relies on open doors and windows to bring fresh air in the wards from outside, which dilutes the concentration of particles such as droplet nuclei containing *Mycobacterium tuberculosis*.

Nurse (N6) said: “the building where TB wards are allocated is old and needs refurbishment. The infrastructure is very old and not user friendly for implementing principles of infection control. Our biggest concern is that the TB wards are perpetuating the spread of infection. The management promised us that they will build new TB wards that are well ventilated, that have good airflow, fans and regularly serviced UV lights as well as big windows and patients’ waiting area to allow airflow. The wards will have nursing station and there will be a demarcation between TB, MDR-TB and XDR-TB. We really need those wards because we are not safe here.”

The findings of this study revealed that nurses need well-ventilated TB wards in the hospital. The TB wards layout was not suitable and lacked ventilation, which increased risk of TB spread. The working environment was not safe and did not meet the national standards guidelines for a TB ward setting, as recommended by South Africa [31]. According to the study by Narasimhan et al. [32], overcrowding and poorly ventilated environments increase the risk of acquiring TB. 

#### 4.6.2. Good Infection Control and Prevention Measures

Good infection control and prevention in the healthcare settings reduce the risk of contracting TB infection. TB settings should have the following TB infection control and prevention measures: workplace and administrative control measures, environmental control measures, and personal control measures. People who work or receive care in healthcare settings are at higher risk of being infected by TB. Therefore, it is necessary to have a TB infection control plan. Healthcare facilities should design policies and procedures for TB control. It should be reviewed periodically and evaluated for effectiveness to determine the actions necessary to minimize the risk for transmission of TB. 

Nurse (N3) indicated that: “We have an urgent need for good quality equipment and personal measures such as N95 respiratory masks, gloves and protective clothes like those in theatres, to protect us from contracting infection with our clothes and expose our families to risks of contracting infection.”

Nurse (N4) said: “We need infection control and prevention to be in place for our safety. It is important that after having contact with patients we should wash our hands with soap. The Provincial government people always say never risk going into TB wards without protective materials.”

The findings of this study revealed that infection control and prevention are not available in TB wards. The nurses expressed a need for those. Nurses and patients were at a great risk of contracting TB infection in the wards. The findings of this study agree with those study by Tshitangano et al. [8], which concluded that infection control plans are not available at the hospitals in Vhembe District, and this results in a high risk of TB infection transmission in the hospitals. 

#### 4.6.3. Special Support from the Management

Nurses play a vital role in the hospital of caring for patients. They are the core members of the hospital who produce good and quality results. Therefore, it is important for the hospital CEO and nursing unit managers to support them in order for them to remain productive. The hospitals should make sure that they order quality equipment, and other materials to assist nurses to do their work effectively. Nurses should be provided with adequate and necessary training to empower them with relevant skills and knowledge. Good management empowers employees in the hospital in different ways and also listens to their requests. Below are the comments made by the participants:

Nurse (N5) said: “Hmmm, we believe that for now we just need special support from the management and all TB programme managers, to empower us with knowledge maybe by having workshops with us or arranging in-service training. It is crucial that we gain more knowledge on how to communicate with patients who are showing side effects or who are likely to default when they are at home. I believe that if we can be empowered with knowledge, the death rate and the spread of disease can be reduced.”

Nurse (N1) said: “I believe that it is important to have meetings with doctors and other management to give us in-service training and workshop.”

According to the findings of this study nurses have various unmet needs when they are doing their job. Participants expressed the need for special support from the management. They were in need of adequate supplies of specialized protective N95 masks, in-service training, workshops, proper equipment, and a daily diet for TB nurses. A study by Sodeify et al. [30] revealed that poor working conditions, lack of resources, special managerial support from the hospitals in South Africa were the reasons why nurses left South Africa to go and work abroad. 

## 5. Limitation of the Study

The study focused on the experience of nurses caring for TB patients at Tshilidzini Hospital in Limpopo Province, South Africa and therefore the findings cannot be generalized. However, according to Smith [33], generalizability in a qualitative study is not intended. The hospital where the study was conducted is situated in the rural area. Therefore, if other hospitals had been included, it could have led to different data findings.

## 6. Recommendations

Based on the above findings, the following recommendations were made: Managers should ensure that TB wards are conducive and adhere to stipulated standards of preventing the spread of infection.Nurses should be provided with necessary equipment and material resources required to provide care to patients in the TB wards.They must implement infection control policies in the wards and health and safety measures must be in place.They must ensure procurement of good quality personal protective materials such as N95 respiratory masks, gloves, and aprons and also follow their usage protocol.Health education should be given to TB patients and their relatives for them to prevent the spread of infection and maintain good hygienic practices.Managers must ensure that they provide their staff with regular in-service training and workshops to equip them with up-to-date information on TB care and infection control practices.Management should provide nurses with debriefing and counselling.Managements should appreciate TB nurses and acknowledge their services through accolades and incentives to boost their morale.TB nurses should be screened for free as stipulated in the Occupational Health and Safety (OHS) Act to monitor their health status

## 7. Conclusions

The study highlighted the plight of nurses when caring for patients suffering from tuberculosis. Nurses indicated their emotional distress due to fear of contagion, lack of material resources, as well as poor infection control practices. The Occupational Health and Safety Act stipulates measures and standards of practice, however such standards are jeopardized by inadequate financial and resource allocation in the public sector. Nurses expressed the need for in-service education and support by managers. Managers must ensure that nurses have resources to maintain quality nursing care and receive social and psychological support as they face death on a daily basis. While available resources are limited, there is a need to direct resources to the problem, to improve quality of services in the TB wards.

## Figures and Tables

**Table 1 ijerph-16-04977-t001:** Themes and sub-themes for the impact of tuberculosis patients on nurses.

Themes	Sub-Themes
1. Challenges in the working environment	1.1: Physical environment
	1.2: Poor positive practice environment
2. Problems impacting on the quality of nursing care	2.1: Adherence to treatment
	2.2: Delay in diagnosis
	2.3: Lack of equipment and working resources
	2.4: Lack of sufficient skills and in-service training
3. Fear, anxiety, stress, and risk of contracting infection	3.1: Exposure to risk
	3.2: Fear of contacting patients
	3.3: Patients’ spreading of disease
4. Nurses perceptions towards patients	4.1: Feelings and emotions of nurses towards the patients
5. Support structure available in the hospital	5.1: Support from the managers
	5.2: Support from the psychologists
	5.3: Appreciation of the staff
6. Support needs for the nurses	6.1: Good ventilated working environment
	6.2: Good infection control and prevention measures
	6.3: Special support from the management

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
