# Peer review of "Caring for Tuberculosis Patients: Understanding the Plight of Nurses at a Regional Hospital in Limpopo Province, South Africa"

_ijerph, 2019, doi:10.3390/ijerph16244977_

Round 1
Reviewer 1 Report
Strengths
---------------------
-Clear, detailed study site description, study population,
sample, methods, data management and analyses
-Summary of themes and subthemes in a table
-Use of details and quotes for themes and subthemes within
section
Weaknesses
---------------------
-Overall, sentence structure and flow is not always clear/correct
Grammar
---------------------
-page 14, line 619 --- sentence should be "pursued in the future"
-page 14, line 620 --- prove is a strong word... maybe use
"provides strong evidence that nurses caring..."
General Questions that would strengthen the conclusion/discussion
----------------------
-How strong is the financial support in this hospital?
-Are there resources to implement any of the recommendations?
-How realistic is it to have any of the recommended changes
completed?
-What kind of public health laws and practices exist? Are they
being violated?
-Is this study designed to draw attention to serious problem
there currently are no resources for? If so, the conclusion
could state that while available resources are limited, there is
a need to direct money and attention to the problem.
Author Response
Please note that this paper was reviewed with the help of the supervisor.
-Overall, sentence structure and flow is not always clear/correct: Note that I have reviewed the paper and worked on the sentence structure and flow.
-page 14, line 619 --- sentence should be "pursued in the future: after a review this line was deleted.
-page 14, line 620 --- prove is a strong word... maybe use
"provides strong evidence that nurses caring... this line was also deleted after a further review.
All this recommendations were attended to:
How strong is the financial support in this hospital? inadequate financial and resource allocation in the public sector is limited.
-Are there resources to implement any of the recommendations? the hospital has limited resources to implement the study recommendations as they depend on the department of health resources allocation.
-How realistic is it to have any of the recommended changes
completed?
-What kind of public health laws and practices exist? Are they
being violated? Occupational health and Safety Act.
-Is this study designed to draw attention to serious problem
there currently are no resources for? If so, the conclusion
could state that while available resources are limited, there is
a need to direct money and attention to the problem.
Reviewer 2 Report
The authors revealed the problems existed in the Tuberculosis patients caring at Tshilidzini hospital in South Africa through in-depth individual interviews. This manuscript is very valuable for other researchers and officials to recognize the harsh realities about Tuberculosis control in Africa. The problems revealed were also concrete and easy to be focused to solve. As to the writing, some suggestions are as follows:
General comments:
Did the interviewer provide an outline for the interviewee in advance? How to evaluate information saturation? Whether information saturation is affected by different investigators? Please clarify. The data management and analysis section needs further explanation. Are there any quality control measures for data collection and analysis? Please discuss how to deal with the problem of infection on purpose for TB grant.
Specific comments:
Line 41, a parenthesis was missing. Line 43, please provide the full name for HCWs. Line 109, could the authors explain the reason why only six participants were interviewed, while there were 22 nurses available? The date of implementation of this project should be specified.Line 239, please specify it is TB grant. Line 264, please delete “they” Is it necessary to use the words “hmmm” “eeeh”? Line 607, please check “DoT”.
Author Response
Did the interviewer provide an outline for the interviewee in advance: The researcher explained the purpose of the study to each participant before interviewing them. Participants were assured that confidentiality will be maintained.
How to evaluate information saturation? Whether information saturation is affected by different investigators? Please clarify. The estimated sample for the interview was fifteen as they met inclusion criteria and ten participants agreed and consented to participate and data saturation was reached with six participants. Interviews were terminated when data saturation was reached, that was when information was repeated and also when the researcher probed, rephrased questions and requested clarity but the information kept repeating itself.
The data management and analysis section needs further explanation. Are there any quality control measures for data collection and analysis? Data was stored in a password-protected computer. Access to the database was restricted to the researcher and supervisors only. Data was stored as per university’s protocols. Any identifiable information that was collected remained confidential and was only accessible to the researcher and supervisors. Data was analyzed in groups not individually to avoid identifying the participants by their responses.
Please discuss how to deal with the problem of infection on purpose for TB grant. TB grant has been cancelled in South Africa since it was causing a problem.
Line 41, a parenthesis was missing. Nurses in high TB burden settings are at higher risk of developing latent tuberculosis infection (LTBI) when compared to the general population, due to their exposure to the large number of smear-positive TB cases managed at hospitals or healthcare facilities.
Line 43, please provide the full name for HCWs. Health Care Workers (HCWs) especially nurses have higher rates of latent and active TB than the general population due to persistent occupational TB exposure, particularly in settings where there is a high prevalence of undiagnosed TB in healthcare facilities and TB infection control programmes are absent or poorly implemented.
Line 109, could the authors explain the reason why only six participants were interviewed, while there were 22 nurses available? The date of implementation of this project should be specified. The estimated sample for the interview was fifteen as they met inclusion criteria and ten participants agreed and consented to participate and data saturation was reached with six participants. Interviews were terminated when data saturation was reached, that was when information was repeated and also when the researcher probed, rephrased questions and requested clarity but the information kept repeating itself.
Line 239, please specify it is TB grant. Nurse (N4) indicated that: ‘’The major problem that we have is that (emotional) our hospital is a rural-based and most patients are from the poor backgrounds; so, they always default their treatment for various reasons, such; continue to receive TB grant,
Line 264, please delete “they” Is it necessary to use the words “hmmm” “eeeh”? they is deleted and all hmm and eeh were deleted and replaced by (emotions). Some of them they default because they go to traditional healers and the healers tell them that they have been bewitched; they are not really sick.
Line 607, please check “DoT”.
Health education should be given to patients and their relatives in order for them prevent the spread of infection and maintain good hygienic practices.
Reviewer 3 Report
Estimado autor
Gracias por su presentación a IJERPH - Las enfermeras que cuidan la tuberculosis son un tema importante para la investigación en salud. Tenga en cuenta las siguientes sugerencias y quizás algunas consultas también para fortalecer este manuscrito.
- El título es informativo, y el resumen es claro, pero este estudio necesita algunas referencias clave, relevantes y recientes de la tuberculosis en cuidados de enfermería.
- Introducción: este es un buen manuscrito explicativo, excepto que algunas oraciones son subjetivas y exponen opiniones de los autores. p 2 línea 64-67. Además, las oraciones necesitan citas.
El objetivo del estudio no es determinar, creo que es explorar.
p2 línea 77 para determinar .... / para explorar;
El objetivo del estudio (pregunta de investigación) se define en la introducción. No obstante, los autores deben indicar con mayor claridad la contribución precisa del trabajo a la literatura existente.
Métodos: me gustaría ver información sobre cómo los participantes fueron reclutados antes en la sección de métodos, ¿es posible proporcionar más detalles sobre las fuentes de reclutamiento para aquellos participantes? Por lo tanto, es importante reconocer la dificultad de reclutar participantes para estudios de este tipo (quizás los autores podrían mencionar esto en la sección de limitaciones). Tamaño de muestra algo pequeño.
El enfoque metodológico utilizado para analizar los datos no es suficiente detalle en el texto.
- Results
The data of participant should presented in an appropriate way.
P4 line 189 one participant said.........
changed for P1: "the hospital infrastructureis very old...
P1 participant 1 or Nurse 1 N1: "....
Another participant said / ... P2 said : or N2: "....
Overall all the results should presented in this way.
The text in the results and the categories re grouped appropriately, but the data in p.12line 522 participants said: "We are working in an old building...
I think Is it repetitive and should be in the first category, or they should explain it differently.
The table is clearly presented, but review the categories.
Los aspectos positivos son que la investigación se realizó de conformidad con los estándares éticos vigentes en el campo.
Los resultados son prácticamente significativos para la necesidad de mejorar el hospital, y se colocan en contexto, y sin ser sobreinterpretados.
El artículo presentado tiene una clara unidad e incluye resultados relevantes en el campo.
Sin embargo, debe revisar las conclusiones y debe responder al objetivo del estudio.
Author Response
P4 line 189 one participant said:
Nurse (N1) said: ’the hospital infrastructure is very old and deteriorating, the wards’ layout is not conducive for TB wards, it has small windows that fuel bacterial concentration.
Nurse (N6) said: ‘’the building where TB wards are allocated is old and needs refurbishment. The infrastructure is very old and not user friendly for implementing principles of infection control. Our biggest concern is that the TB wards are perpetuating the spread of infection
Reviewer 4 Report
The first two words are not MeSH terms.
In the text, reference numbers should be placed in square brackets [ ], and placed before the punctuation; for example [1], [1–3] or [1,3].
Introduction. It is a few long. Must end with prhase The aim of this study was to determine the experiences of nurses caring for TB patients at Tshilidzini Hospital in Limpopo Province, South Africa.
Results are correct.
The reference list should include the full title, as recommended by the ACS style guide
Author Response
In the text, reference numbers should be placed in square brackets [ ], and placed before the punctuation; for example [1], [1–3] or [1,3]. References were done as recommended.
It is estimated that one-third of the world’s population is infected with TB which is responsible for 2-3 million deaths annually [1]. A study conducted by Valjee and van Dyk [2] in South Africa, reported that nurses caring for patients living with TB related illnesses, including HIV/AIDS expressed anxiety, helplessness and vulnerability.
Introduction. It is a few long. Must end with prhase The aim of this study was to determine the experiences of nurses caring for TB patients at Tshilidzini Hospital in Limpopo Province, South Africa.
In addition, shortage of space and beds in the wards hinder separation of MDR-TB and TB patients in healthcare facilities and this increases risk of contracting infection in the wards [10]. Health-care-associated TB has become a major occupational hazard for healthcare workers. It was therefore imperative to explore and describe the experiences of nurses caring for Tuberculosis patients at a regional hospital in Vhembe district, Limpopo Province.
The reference list should include the full title, as recommended by the ACS style guide. it was done as recommended. please see a sample.
Tshitangano, T.G.; Maputle, S.M.; Netshikweta, L.M. Available of Tuberculosis Infection Control Plans at Rural Hospitals of Vhembe District, Limpopo Province of South Africa. African Journal of Primary Health Care & Family Medicine, 2013, 5, 1-6. https://doi.org/10.4102/phcfm.v5i1.480
Round 2
Reviewer 3 Report
Los autores atendieron sugerencias sobre informes y abordaron problemas señalados por mi revisión original. ¡Buen trabajo!
Los autores agregaron al participante como sugirió e informaron sus resultados fortalecieron sus conclusiones.
Su respuesta a los comentarios de los revisores mejoró su manuscrito y su contribución al campo de estudio. Su respuesta fue clara, reflexiva y apropiadas.
Reviewer 4 Report
The reference list should include the full title, as recommended by the ACS style guide. Style files for Endnote and Zotero are available.
The reference list is not correct and it has a major mistakes. We recommend the ACS style guide. Please see a sample and review the references:
1. Author 1, A.B.; Author 2, C.D. Title of the article. Abbreviated Journal Name Year, Volume, page range.
Tshitangano, T.G.; Maputle, S.M.; Netshikweta, L.M. Available of Tuberculosis Infection Control Plans at Rural Hospitals of Vhembe District, Limpopo Province of South Africa. African Journal of Primary Health Care & Family Medicine, 2013, 5, 1-6. https://doi.org/10.4102/phcfm.v5i1.480